# Joint Autoregressive and Hierarchical Priors for Learned Image Compression

**David Minnen, Johannes Ballé, George Toderici**
Google Research
{dminnen, jballe, gtoderici}@google.com

## Abstract

Recent models for learned image compression are based on autoencoders that learn approximately invertible mappings from pixels to a quantized latent representation. The transforms are combined with an entropy model, which is a prior on the latent representation that can be used with standard arithmetic coding algorithms to generate a compressed bitstream. Recently, hierarchical entropy models were introduced as a way to exploit more structure in the latents than previous fully factorized priors, improving compression performance while maintaining end-to-end optimization. Inspired by the success of autoregressive priors in probabilistic generative models, we examine autoregressive, hierarchical, and combined priors as alternatives, weighing their costs and benefits in the context of image compression. While it is well known that autoregressive models can incur a significant computational penalty, we find that in terms of compression performance, autoregressive and hierarchical priors are complementary and can be combined to exploit the probabilistic structure in the latents better than all previous learned models. The combined model yields state-of-the-art rate–distortion performance and generates smaller files than existing methods: 15.8% rate reductions over the baseline hierarchical model and 59.8%, 35%, and 8.4% savings over JPEG, JPEG2000, and BPG, respectively. To the best of our knowledge, our model is the first learning-based method to outperform the top standard image codec (BPG) on both the PSNR and MS-SSIM distortion metrics.

## 1   Introduction

Most recent methods for learning-based, lossy image compression adopt an approach based on *transform coding* [1]. In this approach, image compression is achieved by first mapping pixel data into a quantized latent representation and then losslessly compressing the latents. Within the deep learning research community, the transforms typically take the form of convolutional neural networks (CNNs), which learn nonlinear functions with the potential to map pixels into a more compressible latent space than the linear transforms used by traditional image codecs. This nonlinear transform coding method resembles an autoencoder [2], [3], which consists of an *encoder* transform from the data (in this case, pixels) to a reduced-dimensionality latent space, and a *decoder*, an approximate inverse function that maps latents back to pixels. While dimensionality reduction can be seen as a simple form of compression, it is not equivalent since the goal of compression is to reduce the entropy of the representation under a prior probability model shared between the sender and the receiver (the *entropy model*), not just to reduce the dimensionality. To improve compression performance, recent methods have focused on better encoder/decoder transforms and on more sophisticated entropy models [4]–[14]. Finally, the entropy model is used in conjunction with standard entropy coding algorithms such as arithmetic, range, or Huffman coding [15]–[17] to generate a compressed bitstream.

The training goal is to minimize the expected length of the bitstream as well as the expected distortion of the reconstructed image with respect to the original, giving rise to a rate–distortion optimization problem:

$$R + \lambda \cdot D = \underbrace{\mathbb{E}_{\boldsymbol{x} \sim p_{\boldsymbol{x}}}\big[-\log_2 p_{\hat{\boldsymbol{y}}}(\lfloor f(\boldsymbol{x})\rceil)\big]}_{\text{rate}} + \lambda \cdot \underbrace{\mathbb{E}_{\boldsymbol{x} \sim p_{\boldsymbol{x}}}\big[d(\boldsymbol{x}, g(\lfloor f(\boldsymbol{x})\rceil))\big]}_{\text{distortion}}, \qquad (1)$$

where $\lambda$ is the Lagrange multiplier that determines the desired rate–distortion trade-off, $p_{\boldsymbol{x}}$ is the unknown distribution of natural images, $\lfloor \cdot \rceil$ represents rounding to the nearest integer (quantization), $\boldsymbol{y} = f(\boldsymbol{x})$ is the encoder, $\hat{\boldsymbol{y}} = \lfloor \boldsymbol{y} \rceil$ are the quantized latents, $p_{\hat{\boldsymbol{y}}}$ is a discrete entropy model, and $\hat{\boldsymbol{x}} = g(\hat{\boldsymbol{y}})$ is the decoder with $\hat{\boldsymbol{x}}$ representing the reconstructed image. The rate term corresponds to the cross entropy between the marginal distribution of the latents and the learned entropy model, which is minimized when the two distributions are identical. The distortion term may correspond to a closed-form likelihood, such as when $d(\boldsymbol{x}, \hat{\boldsymbol{x}})$ represents mean squared error (MSE), which allows the model to be interpreted as a variational autoencoder [5], [6]. When optimizing the model for other distortion metrics (e.g., SSIM or MS-SSIM), it is simply minimized as an energy function.

The models we analyze in this paper build on the work of Ballé et al. [13], which uses a noise-based relaxation to apply gradient descent methods to the loss function in Eq. (1) and which introduces a hierarchical prior to improve the entropy model. While most previous research uses a fixed, though potentially complex, entropy model, Ballé et al. use a Gaussian scale mixture (GSM) [18] where the scale parameters are conditioned on a hyperprior. Their model allows for end-to-end training, which includes joint optimization of a quantized representation of the hyperprior, the conditional entropy model, and the base autoencoder. The key insight of their work is that the compressed hyperprior can be added to the generated bitstream as *side information*, which allows the decoder to use the conditional entropy model. In this way, the entropy model itself is image-dependent and spatially adaptive, which allows for a richer and more accurate model. Ballé et al. show that standard optimization methods for deep neural networks are sufficient to learn a useful balance between the size of the side information and the savings gained from a more accurate entropy model. The resulting compression model provides state-of-the-art image compression results compared to earlier learning-based methods.

We extend this GSM-based entropy model in two ways: first, by generalizing the hierarchical GSM model to a Gaussian mixture model (GMM), and, inspired by recent work on generative models, by adding an autoregressive component. We assess the compression performance of both approaches, including variations in the network architectures, and discuss benefits and potential drawbacks of both extensions. For the results in this paper, we did not investigate the effect of reducing the capacity (i.e., the number of layers and number of channels) of the deep networks to optimize computational complexity, since we are interested in determining the potential of different forms of priors rather than trading off complexity against performance. Note that increasing capacity alone is not sufficient to obtain arbitrarily good compression performance [13, appendix 6.3].

## 2  Architecture Details

Figure 1 provides a high-level overview of our generalized compression model, which contains two main sub-networks[1]. The first is the core autoencoder, which learns a quantized latent representation of images (*Encoder* and *Decoder* blocks). The second sub-network is responsible for learning a probabilistic model over quantized latents used for entropy coding. It combines the *Context Model*, an autoregressive model over latents, with the hyper-network (*Hyper Encoder* and *Hyper Decoder* blocks), which learns to represent information useful for correcting the context-based predictions. The data from these two sources is combined by the *Entropy Parameters* network, which generates the mean and scale parameters for a conditional Gaussian entropy model.

Once training is complete, a valid compression model must prevent any information from passing between the encoder to the decoder unless that information is available in the compressed file. In Figure 1, the arithmetic encoding (AE) blocks produce the compressed representation of the symbols coming from the quantizer, which is stored in a file. Therefore at decoding time, any information that depends on the quantized latents may be used by the decoder once it has been decoded. In order for

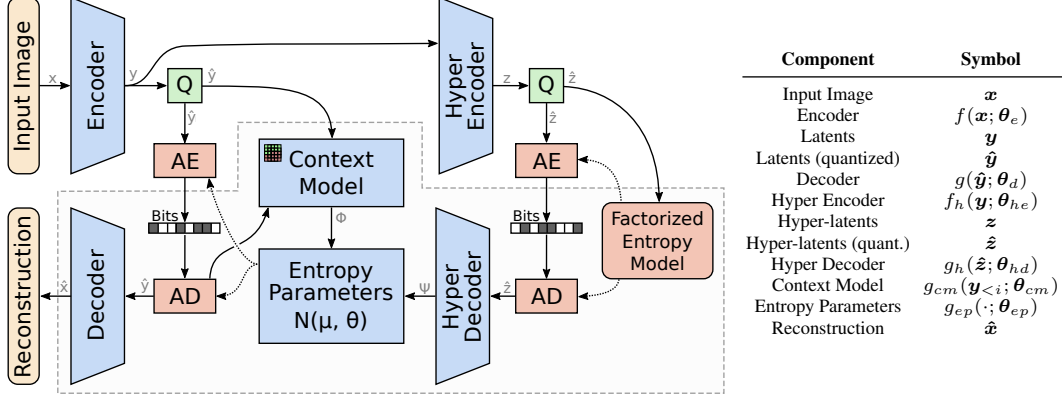

Figure 1: Our combined model jointly optimizes an autoregressive component that predicts latents from their causal context (*Context Model*) along with a hyperprior and the underlying autoencoder. Real-valued latent representations are quantized ($Q$) to create integer-valued latents ($\hat{y}$) and hyper-latents ($\hat{z}$), which are compressed into a bitstream using an arithmetic encoder (*AE*) and decompressed by an arithmetic decoder (*AD*). The highlighted region corresponds to the components that are executed by the receiver to recover an image from a compressed bitstream.

the context model to work, at any point it can only access the latents that have already been decoded. When starting to decode an image, we assume that the previously decoded latents have all been set to zero.

The learning problem is to minimize the expected rate–distortion loss defined in Eq. 1 over the model parameters. Following the work of Ballé et al. [13], we model each latent, $\hat{y}_i$, as a Gaussian convolved with a unit uniform distribution. This ensures a good match between encoder and decoder distributions of both the quantized latents, and continuous-valued latents subjected to additive uniform noise during training. While [13] predicted the scale of each Gaussian conditioned on the hyperprior, $\hat{z}$, we extend the model by predicting the mean and scale parameters conditioned on both the hyperprior as well as the causal context of each latent $\hat{y}_i$, which we denote by $\hat{y}_{<i}$. The predicted Gaussian parameters are functions of the learned parameters of the hyper-decoder, context model, and entropy parameters networks ($\boldsymbol{\theta}_{hd}$, $\boldsymbol{\theta}_{cm}$, and $\boldsymbol{\theta}_{ep}$, respectively):

$$p_{\hat{\boldsymbol{y}}}(\hat{\boldsymbol{y}} \mid \hat{\boldsymbol{z}}, \boldsymbol{\theta}_{hd}, \boldsymbol{\theta}_{cm}, \boldsymbol{\theta}_{ep}) = \prod_i \Big( \mathcal{N}\big(\mu_i, \sigma_i^2\big) * \mathcal{U}\big(-\tfrac{1}{2}, \tfrac{1}{2}\big) \Big)(\hat{y}_i)$$

$$\text{with } \mu_i, \sigma_i = g_{ep}(\boldsymbol{\psi}, \boldsymbol{\phi}_i; \boldsymbol{\theta}_{ep}), \boldsymbol{\psi} = g_h(\hat{\boldsymbol{z}}; \boldsymbol{\theta}_{hd}), \text{and } \boldsymbol{\phi}_i = g_{cm}(\hat{\boldsymbol{y}}_{<i}; \boldsymbol{\theta}_{cm}). \quad (2)$$

The entropy model for the hyperprior is the same as in [13], although we expect the hyper-encoder and hyper-decoder to learn significantly different functions in our combined model, since they now work in conjunction with an autoregressive network to predict the parameters of the entropy model. Since we do not make any assumptions about the distribution of the hyper-latents, a non-parametric, fully factorized density model is used. A more powerful entropy model for the hyper-latents may improve compression rates, e.g., we could stack multiple instances of our contextual model, but we expect the net effect to be minimal since, empirically, $\hat{z}$ comprises only a very small percentage of the total file size. Because both the compressed latents and the compressed hyper-latents are part of the generated bitstream, the rate–distortion loss from Equation 1 must be expanded to include the cost of transmitting $\hat{z}$. Coupled with a squared error distortion metric, the full loss function becomes:

$$R + \lambda \cdot D = \underbrace{\mathbb{E}_{\boldsymbol{x} \sim p_{\boldsymbol{x}}}\big[-\log_2 p_{\hat{\boldsymbol{y}}}(\hat{\boldsymbol{y}})\big]}_{\text{rate (latents)}} + \underbrace{\mathbb{E}_{\boldsymbol{x} \sim p_{\boldsymbol{x}}}\big[-\log_2 p_{\hat{\boldsymbol{z}}}(\hat{\boldsymbol{z}})\big]}_{\text{rate (hyper-latents)}} + \lambda \cdot \underbrace{\mathbb{E}_{\boldsymbol{x} \sim p_{\boldsymbol{x}}}\|\boldsymbol{x} - \hat{\boldsymbol{x}}\|_2^2}_{\text{distortion}} \quad (3)$$

## 2.1 Layer Details and Constraints

Details about the individual network layers in each component of our models are outlined in Table 1. While the internal structure of the components is fairly unrestricted, e.g., one could exchange the convolutional layers for residual blocks or dilated convolution without fundamentally changing the

| Encoder | Decoder | Hyper Encoder | Hyper Decoder | Context Prediction | Entropy Parameters |
|---|---|---|---|---|---|
| Conv: 5×5 c192 s2 | Deconv: 5×5 c192 s2 | Conv: 3×3 c192 s1 | Deconv: 5×5 c192 s2 | Masked: 5×5 c384 s1 | Conv: 1×1 c640 s1 |
| GDN | IGDN | Leaky ReLU | Leaky ReLU | | Leaky ReLU |
| Conv: 5×5 c192 s2 | Deconv: 5×5 c192 s2 | Conv: 5×5 c288 s2 | Deconv: 5×5 c288 s2 | | Conv: 1×1 c512 s1 |
| GDN | IGDN | Leaky ReLU | Leaky ReLU | | Leaky ReLU |
| Conv: 5×5 c192 s2 | Deconv: 5×5 c192 s2 | Conv: 5×5 c192 s2 | Deconv: 3×3 c384 s1 | | Conv: 1×1 c384 s1 |
| GDN | IGDN | | | | |
| Conv: 5×5 c192 s2 | Deconv: 5×5 c3 s2 | | | | |

Table 1: Each row corresponds to a layer of our generalized model. Convolutional layers are specified with the "Conv" prefix followed by the kernel size, number of channels and downsampling stride (e.g., the first layer of the encoder uses 5×5 kernels with 192 channels and a stride of two). The "Deconv" prefix corresponds to upsampled convolutions (i.e., in TensorFlow, `tf.conv2d_transpose`), while "Masked" corresponds to masked convolution as in [19]. GDN stands for generalized divisive normalization, and IGDN is inverse GDN [20].

model, certain components must be constrained to ensure that the bitstreams alone is sufficient for the receiver to reconstruct the image.

The last layer of the encoder corresponds to the bottleneck of the base autoencoder. The number of output channels determines the number of elements that must be compressed and stored. Depending on the rate–distortion trade-off, our models learn to ignore certain channels by deterministically generating the same latent value and assigning it a probability of 1, which wastes computation but generates no additional entropy. This modeling flexibility allows us to set the bottleneck larger than necessary, and then let the model determine the number of channels that yields the best performance. Similar to reports in other work, we found that too few channels can impede rate–distortion performance when training models that target high bit rates, but having too many does not harm the compression performance [9], [13].

The final layer of the decoder must have three channels to generate RGB images, and the final layer of the *Entropy Parameters* sub-network must have exactly twice as many channels as the bottleneck. This constraint arises because the *Entropy Parameters* network predicts two values, the mean and scale of a Gaussian distribution, for each latent. The number of output channels of the *Context Model* and *Hyper Decoder* components are not constrained, but we also set them to twice the bottleneck size in all of our experiments.

Although the formal definition of our model allows the autoregressive component to condition its predictions $\phi_i = g_{cm}(\hat{\boldsymbol{y}}_{<i}; \boldsymbol{\theta}_{cm})$ on all previous latents, in practice we use a limited context (5×5 convolution kernels) with masked convolution similar to the approach used by PixelCNN [19]. The *Entropy Parameters* network is also constrained, since it can not access predictions from the *Context Model* beyond the current latent element. For simplicity, we use 1×1 convolution in the *Entropy Parameters* network, although masked convolution is also permissible. Section 3 provides an empirical evaluation of the model variants we assessed, exploring the effects of different context sizes and more complex autoregressive networks.

## 3 Experimental Results

We evaluate our generalized models by calculating the rate–distortion (RD) performance averaged over the publicly available Kodak image set [21][2]. Figure 2 shows RD curves using peak signal-to-noise ratio (PSNR) as the image quality metric. While PSNR is known to be a relatively poor perceptual metric [22], it is still a standard metric used to evaluate image compression algorithms and is the primary metric used for tuning conventional codecs. The RD graph on the left of Figure 2 compares our combined context + hyperprior model to existing image codecs (standard codecs and learned models) and shows that this model outperforms all of the existing methods including BPG [23], a state-of-the-art codec based on the intra-frame coding algorithm from HEVC [24]. To the best of our knowledge, this is the first learning-based compression model to outperform BPG on

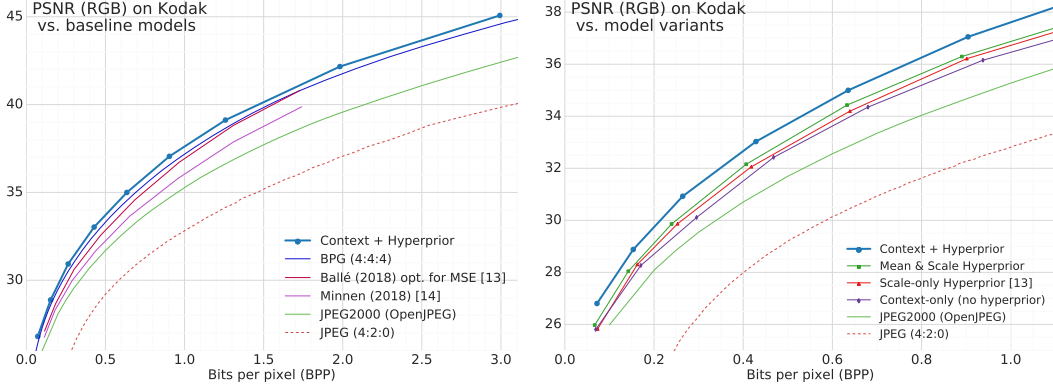

Figure 2: Our combined approach (context + hyperprior) has better rate–distortion performance on the Kodak image set as measured by PSNR (RGB) compared to all of the baselines methods (*left*). To our knowledge, this is the first learning-based method to outperform BPG on PSNR. The right graph compares the relative performance of different versions of our method. It shows that using a hyperprior is better than a purely autoregressive (context-only) approach and that combining both (context + hyperprior) yields the best RD performance.

PSNR. The right RD graph compares different versions of our models and shows that the combined model performs the best, while the context-only model performs slightly worse than either hierachical version.

Figure 3 shows RD curves for Kodak using multiscale structural similarity (MS-SSIM) [25] as the image quality metric. The graph includes two versions of our combined model: one optimized for MSE and one optimized for MS-SSIM. The latter outperforms all existing methods including all standard codecs and other learning-based methods that were also optimized for MS-SSIM ([6], [9], [13]). As expected, when our model is optimized for MSE, performance according to MS-SSIM falls. Nonetheless, the MS-SSIM scores for this model still exceed all standard codecs and all learning-based methods that were not specifically optimized for MS-SSIM.

As outlined in Table 1, our baseline architecture for the combined model uses $5 \times 5$ masked convolution in a single linear layer for the context model, and it uses a conditional Gaussian distribution for the entropy model. Figure 4 compares this baseline to several variants by showing the relative increase in file size at a single rate-point. The green bars show that exchanging the Gaussian distribution for a logistic distribution has almost no effect (the 0.3% increase is smaller than the training variance), while switching to a Laplacian distribution decreases performance more substantially. The blue bars compare different context configurations. Masked $3 \times 3$ and $7 \times 7$ convolution both perform slightly worse, which is surprising since we expected the additional context provided by the $7 \times 7$ kernels to improve prediction accuracy. Similarly, a 3-layer, nonlinear context model using $5 \times 5$ masked convolution also performed slightly worse than the linear baseline. Finally, the purple bars show the effect of using a severely restricted context such as only a single neighbor or three neighbors from the previous row. The primary benefit of these models is increased parallelization when calculating context-based predictions since the dependence is reduced from two dimensions down to one. While both cases show a non-negligible rate increase (2.1% and 3.1%, respectively), the increase may be worthwhile in a practical implementation where runtime speed is a major concern.

Finally, Figure 5 provides a visual comparison for one of the Kodak images. Creating accurate comparisons is difficult since most compression methods do not have the ability to target a precise bit rate. We therefore selected comparison images with sizes that are as close as possible, but always larger than our encoding (up to 9.4% larger in the case of BPG). Nonetheless, our compression model provides clearly better visual quality compared to the scale hyperprior baseline [13] and JPEG. The perceptual quality relative to BPG is much closer. For example, BPG preserves more detail in the sky and parts of the fence, but at the expense of introducing geometric artifacts in the sky, mild ringing near the building/sky boundaries, and some boundary artifacts where neighboring blocks have widely different levels of detail (e.g., in the grass and lighthouse).

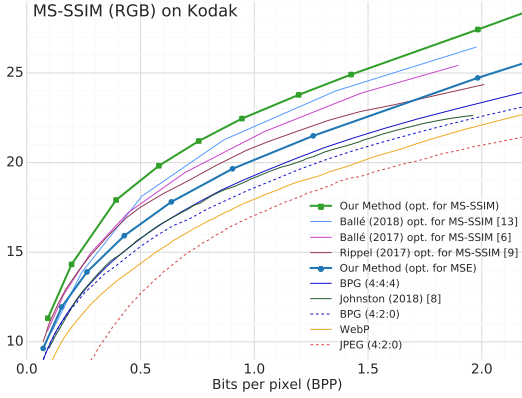

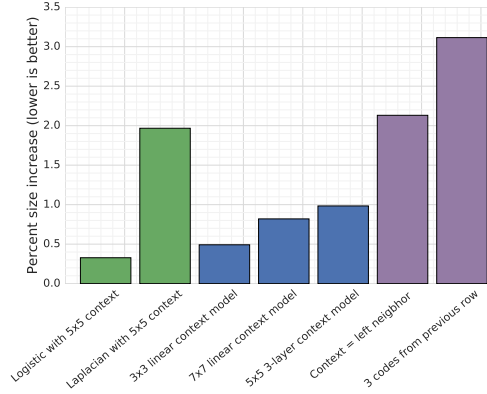

Figure 3: When evaluated using MS-SSIM (RGB) on Kodak, our combined approach has better RD performance than all previous methods when optimized for MS-SSIM. When optimized for MSE, our method still provides better MS-SSIM scores than all of the standard codecs.

Figure 4: The baseline implementation of our model uses a hyperprior and a linear context model with $5{\times}5$ masked convolution. Optimized with $\lambda = 0.025$ (bpp $\approx 0.61$ on Kodak), the baseline outperforms the other variants we tested (see text for details).

## 4 Related Work

The earliest research that used neural networks to compress images dates back to the 1980s and relies on an autoencoder with a small bottleneck using either uniform quantization [26] or vector quantization [27], [28]. These approaches sought equal utilization of the codes and thus did not learn an explicit entropy model. Considerable research followed these initial models, and Jiang provides a comprehensive survey covering methods published through the late 1990s [29].

More recently, image compression with deep neural networks became a popular research topic starting with the work of Toderici et al. [30] who used a recurrent architecture based on LSTMs to learn multi-rate, progressive models. Their approach was improved by exploring other recurrent architectures for the autoencoder, training an LSTM-based entropy model, and adding a post-process that spatially adapts the bit rate based on the complexity of the local image content [4], [8]. Related

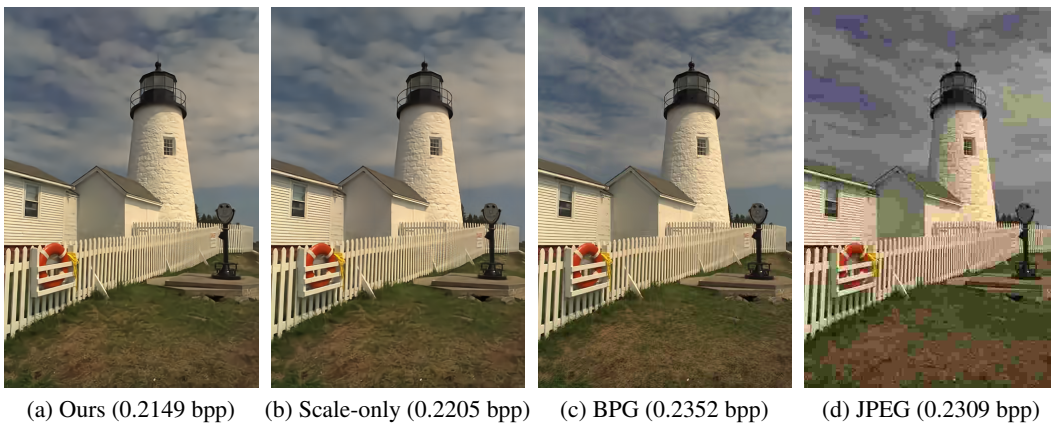

(a) Ours (0.2149 bpp)    (b) Scale-only (0.2205 bpp)    (c) BPG (0.2352 bpp)    (d) JPEG (0.2309 bpp)

Figure 5: At similar bit rates, our combined method provides the highest visual quality. Note the aliasing in the fence in the scale-only version as well as a slight global color cast and blurriness in the yellow rope. BPG shows more "classical" compression artifacts, e.g., ringing around the top of the lighthouse and the roof of the middle building. BPG also introduces a few geometric artifacts in the sky, though it does preserve more detail in the sky and fence compared to our model, albeit with 9.4% more bits. JPEG shows severe blocking artifacts at this bit rate.

research followed a more traditional image coding approach and explicitly divided images into patches instead of using a fully convolutional model [10], [31]. Inspired by modern image codecs and learned inpainting algorithms, these methods trained a neural network to predict each image patch from its causal context (in the image, not the latent space) before encoding the residual. Similarly, most modern image compression standards use context to predict pixel values combined with a context-adaptive entropy model [23], [32], [33].

Many learning-based methods take the form of an autoencoder, and separate models are trained to target different bit rates instead of training a single recurrent model [5]–[7], [9], [11], [12], [14], [34], [35]. Some use a fully factorized entropy model [5], [6], while others make use of context in code space to improve compression rates [4], [7]–[9], [12], [35]. Other methods do not make use of context via an autoregressive model and instead rely on side information that is either predicted by a neural network [13] or composed of indices into a (shared) dictionary of non-parametric code distributions used locally by the entropy coder [14]. In concurrent research, Klopp et al. also explore an approach that jointly optimizes a context model and a hierarchical prior [35]. They introduce a sparse variant of GDN to improve the encoder and decoder networks and use a multimodal entropy distribution. Their integration method between the context model and hyperprior is somewhat simpler than our approach, which leads to a final model with slightly worse rate–distortion performance (~10% higher bit rates for equivalent MS-SSIM).

Learned image compression is also related to Bayesian generative models such as PixelCNN [19], variational autoencoders [36], PixelVAE [37], $\beta$-VAE [38], and VLAE [39]. In general, Bayesian image models seek to maximize the *evidence* $\mathbb{E}_{\boldsymbol{x} \sim p_{\boldsymbol{x}}} \log p(\boldsymbol{x})$, which is typically intractable, and use the joint likelihood, as in Eq. (1), as a lower bound, while compression models seek to directly optimize Eq. (1). It has been noted that under certain conditions, compression models are formally equivalent to VAEs [5], [6]. $\beta$-VAEs have a particularly strong connection since $\beta$ controls the trade-off between the data log-likelihood (distortion) and prior (rate), as does $\lambda$ in our formulation, which is derived from classical rate–distortion theory.

Another significant difference are the constraints imposed on compression models by the need to quantize and arithmetically encode the latents, which require certain choices regarding the parametric form of the densities and a transition from continuous (differential) to discrete (Shannon) entropies. We can draw strong conceptual parallels between our models and PixelCNN autoencoders [19], and especially PixelVAE [37] and VLAE [39], when applied to discrete latents. These models are often evaluated by comparing average likelihoods (which correspond to differential entropies), whereas compression models are typically evaluated by comparing several bit rates (corresponding to Shannon entropies) and distortion values across the rate–distortion frontier, which makes direct comparison more complex.

## 5    Discussion

Our approach extends the work of Ballé et al. [13] in two ways. First, we generalize the GSM model to a conditional Gaussian mixture model (GMM). Supporting this model is simply a matter of generating both a mean and a scale parameter conditioned on the hyperprior. Intuitively, the average likelihood of the observed latents increases when the center of the conditional Gaussian is closer to the true value and a smaller scale is predicted, i.e., more structure can be exploited by modeling conditional means. The core question is whether or not the benefits of this more sophisticated model outweigh the cost of the associated side information. We showed in Figure 2 (*right*) that a GMM-based entropy model provides a net benefit and outperforms the simpler GSM-based model in terms of rate–distortion performance without increasing the asymptotic complexity of the model.

The second extension is the idea of combining an autoregressive model with the hyperprior. Intuitively, we can see how these components are complementary in two ways. First, starting from the perspective of the hyperprior, we see that for identical hyper-network architectures, improvements to the entropy model require more side information. The side information increases the total compressed file size, which limits its benefit. In contrast, introducing an autoregressive component into the prior does not incur a rate penalty since the predictions are based only on the causal context, i.e., on latents that have already been decoded. Similarly, from the perspective of the autoregressive model, we expect some amount of uncertainty that can not be eliminated solely from the causal context. The hyperprior, however, can "look into the future" since it is part of the compressed bitstream and is fully known by

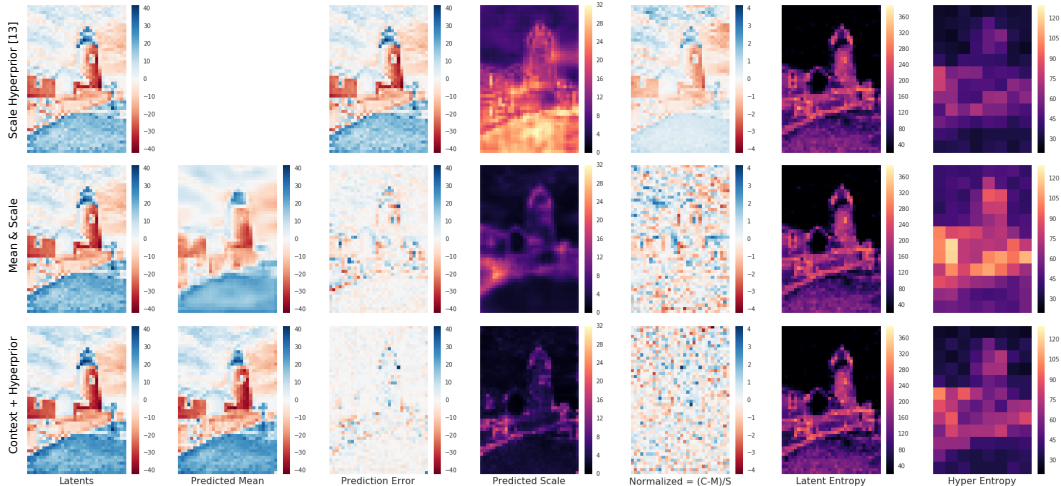

Figure 6: Each row corresponds to a different model variant and shows information for the channel with the highest entropy. The visualizations show that more powerful models reduce the prediction error, require smaller scale parameters, and remove structure from the normalized latents, which directly translates into a more accurate entropy model and thus higher compression rates.

the decoder. The hyperprior can thus learn to store information needed to reduce the uncertainty in the autoregressive model while avoiding information that can be accurately predicted from context.

Figure 6 visualizes some of the internal mechanisms of our models. We show three of the variants: one Gaussian scale mixture equivalent to [13], another strictly hierarchical prior extended to a Gaussian mixture model, and one combined model using an autoregressive component and a hyperprior. After encoding the lighthouse image shown in Figure 5, we extracted the latents for the channel with the highest entropy. These latents are visualized in the first column of Figure 6. The second column holds the conditional means and clearly shows the added detail attained with an autoregressive component, which is reminiscent of the observation that VAE-based models tend to produce blurrier images than autoregressive models [37]. This improvement leads to a lower prediction error (third column) and smaller predicted scales, i.e. smaller uncertainty (fourth column). Our entropy model assumes that latents are conditionally independent given the hyperprior, which implies that the normalized latents, i.e. values with the predicted mean and scale removed, should be closer to i.i.d. Gaussian noise. The fifth column of Figure 6 shows that the combined model is closest to this ideal and that both the mean prediction and autoregressive model help significantly. Finally, the last two columns show how the entropy is distributed across the image for the latents and hyper-latents.

From a practical standpoint, autoregressive models are less desirable than hierarchical models since they are inherently serial, and therefore can not be sped up using techniques such as parallelization. To report the rate–distortion performance of the compression models which contain an autoregressive component, we refrained from implementing a full decoder for this paper, and instead compare Shannon entropies. We have empirically verified that these measurements are within a fraction of a percent of the size of the actual bitstream generated by arithmetic coding.

Probability density distillation has been successfully used to get around the serial nature of autoregressive models for the task of speech synthesis [40], but unfortunately the same type of method cannot be applied in the domain of compression due to the coupling between the prior and the arithmetic decoder. To address these computational concerns, we have begun to explore very lightweight context models as described in Section 3 and Figure 4, and are considering further techniques to reduce the computational requirements of the *Context Model* and *Entropy Parameters* networks, such as engineering a tight integration of the arithmetic decoder with a differentiable autoregressive model. An alternative direction for future research may be to avoid the causality issue altogether by introducing yet more complexity into strictly hierarchical priors or adopt an interleaved decomposition for context prediction that allows partial parallelization [34], [41].

## Footnotes

[1]See Section 4 in the supplemental materials for an in-depth visual comparison between our architecture variants and previous learning-based methods.

[2]Please see the supplemental material for additional evaluation results including full-page RD curves, example images, and results on the larger Tecnick image set (100 images with resolution 1200×1200).

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
