[Reviews · NeurIPS 2018]

Reviewer 1



The paper describes a novel model for image compression that uses neural networks. The novel model extends the existing Ballé's (2018) model by adding two elements: (1) Generalizing the hierarchical GSM model to a Gaussian mixture model (2) adding an autoregressive component. As the result section describes, the model seems to be effective on the Kodak Dataset. In fact, it outperforms Ballé's method as well as variant of this method. Overall, the paper seems to be a solid paper with interesting results. However, the authors do not help in understanding how the model is in fact useful in pratical situations or why it is important from the point of view of its theorical results.

Reviewer 2



This paper extends the work of Ballé et al. from GSM model to GMM model, and also added an autoregressive model with the hyper-prior. By doing that the paper reached a better performance compared to baseline approaches including Balle et al. As I don't have deep understanding of the context and the previous literature, I can't judge too much of the technical quality here.

Reviewer 3



Summary ======= This paper extends the autoencoder trained for compression of Balle et al. (2018) with a small autoregressive model. The autoencoder of Balle uses Gaussian scale mixtures (GSMs) for entropy encoding of coefficients, and encodes its latent variables as side information in the bit stream. Here, conditional Gaussian mixtures are used which additionally use neighboring coefficients as context. The authors find that this significantly improves compression performance. Good ==== – Good performance (notably, state-of-the-art MS-SSIM results without optimizing directly on this metric) – Extensive supplementary materials, including rate-distortion curves for individual images – Well written Bad === – Incremental, with no real conceptual contributions – Missing related work: There is a long history of conditional Gaussian mixture models for autoregressive modeling of images – including for entropy rate estimation – that is arguably more relevant than other generative models mentioned in the paper: Domke et al. (2008), Hosseini et al. (2010), Theis et al. (2012), Uria et al. (2013), Theis et al. (2015)